# Empirical Analysis of Unlabeled Entity Problem in Named Entity Recognition

**Yangming Li, Lemao Liu, & Shuming Shi**
Tencent AI Lab
{newmanli,redmondliu,shumingshi}@tencent.com

## Abstract

In many scenarios, named entity recognition (NER) models severely suffer from *unlabeled entity problem*, where the entities of a sentence may not be fully annotated. Through empirical studies performed on synthetic datasets, we find two causes of performance degradation. One is the reduction of annotated entities and the other is treating unlabeled entities as negative instances. The first cause has less impact than the second one and can be mitigated by adopting pretraining language models. The second cause seriously misguides a model in training and greatly affects its performances. Based on the above observations, we propose a general approach, which can almost eliminate the misguidance brought by unlabeled entities. The key idea is to use negative sampling that, to a large extent, avoids training NER models with unlabeled entities. Experiments on synthetic datasets and real-world datasets show that our model is robust to *unlabeled entity problem* and surpasses prior baselines. On well-annotated datasets, our model is competitive with the state-of-the-art method[1].

## 1 Introduction

Named entity recognition (NER) is an important task in information extraction. Previous methods typically cast it into a sequence labeling problem by adopting IOB tagging scheme (Mesnil et al., 2015; Huang et al., 2015; Ma & Hovy, 2016; Akbik et al., 2018; Qin et al., 2019). A representative model is Bi-LSTM CRF (Lample et al., 2016). The great success achieved by these methods benefits from massive correctly labeled data. However, in some real scenarios, not all the entities in the training corpus are annotated. For example, in some NER tasks (Ling & Weld, 2012), the datasets contain too many entity types or a mention may be associated with multiple labels. Since manual annotation on this condition is too hard, some entities are inevitably neglected by human annotators. Situations in distantly supervised NER (Ren et al., 2015; Fries et al., 2017) are even more serious. To reduce handcraft annotation, distant supervision (Mintz et al., 2009) is applied to automatically produce labeled data. As a result, large amounts of entities in the corpus are missed due to the limited coverage of knowledge resources. We refer this to *unlabeled entity problem*, which largely degrades performances of NER models.

There are several approaches used in prior works to alleviate this problem. Fuzzy CRF and AutoNER (Shang et al., 2018b) allow models to learn from the phrases that may be potential entities. However, since these phrases are obtained through a distantly supervised phrase mining method (Shang et al., 2018a), many unlabeled entities in the training data may still not be recalled. In the context of only resorting to unlabeled corpora and an entity ontology, Mayhew et al. (2019); Peng et al. (2019) employ positive-unlabeled (PU) learning (Li & Liu, 2005) to unbiasedly and consistently estimate the task loss. In implementations, they build distinct binary classifiers for different labels. Nevertheless, the unlabeled entities still impact the classifiers of the corresponding entity types and, importantly, the model can't disambiguate neighboring entities. Partial CRF (Tsuboi et al., 2008) is an extension of commonly used CRF (Lafferty et al., 2001) that supports learning from incomplete annotations. Yang et al. (2018); Nooralahzadeh et al. (2019); Jie et al. (2019) use it to circumvent training with false negatives. However, as fully annotated corpora are still required

---

[1]Our source code is available at https://github.com/LeePleased/NegSampling-NER.

to get ground truth training negatives, this approach is not applicable to the situations where little or even no high-quality data is available.

In this work, our goal is to study what are the impacts of *unlabeled entity problem* on the models and how to effectively eliminate them. Initially, we construct some synthetic datasets and introduce degradation rates. The datasets are constructed by randomly removing the annotated named entities in well-annotated datasets, e.g., CoNLL-2003 (Sang & De Meulder, 2003), with different probabilities. The degradation rates measure how severe an impact of *unlabeled entity problem* degrades the performances of models. Extensive studies are investigated on synthetic datasets. We find two causes: the reduction of annotated entities and treating unlabeled entities as negative instances. The first cause is obvious but has far fewer influences than the second one. Besides, it can be mitigated well by using a pretraining language model, like BERT (Devlin et al., 2019)), as the sentence encoder. The second cause seriously misleads the models in training and exerts a great negative impact on their performances. Even in less severe cases, it can sharply reduce the F1 score by about $20\%$. Based on the above observations, we propose a novel method that is capable of eliminating the misguidance of unlabeled entities in training. The core idea is to apply negative sampling that avoids training NER models with unlabeled entities.

Extensive experiments have been conducted to verify the effectiveness of our approach. Studies on synthetic datasets and real-world datasets (e.g., EC) show that our model well handles unlabeled entities and notably surpasses prior baselines. On well-annotated datasets (e.g., CoNLL-2003), our model is competitive with the state-of-the-art method.

## 2 PRELIMINARIES

In this section, we formally define the *unlabeled entity problem* and briefly describe a strong baseline, BERT Tagging (Devlin et al., 2019), used in empirical studies.

### 2.1 UNLABELED ENTITY PROBLEM

We denote an input sentence as $\mathbf{x} = [x_1, x_2, \cdots, x_n]$ and the annotated named entity set as $\mathbf{y} = \{y_1, y_2, \cdots, y_m\}$. $n$ is the sentence length and $m$ is the amount of entities. Each member $y_k$ of set $\mathbf{y}$ is a tuple $(i_k, j_k, l_k)$. $(i_k, j_k)$ is the span of an entity which corresponds to the phrase $\mathbf{x}_{i_k, j_k} = [x_{i_k}, x_{i_k+1}, \cdots, x_{j_k}]$ and $l_k$ is its label. The *unlabeled entity problem* is defined as, due to the limited coverage of machine annotator or the negligence of human annotator, some ground truth entities $\widetilde{\mathbf{y}}$ of the sentence $\mathbf{x}$ are not covered by annotated entity set $\mathbf{y}$.

For instance, given a sentence $\mathbf{x} = [\text{Jack}, \text{and}, \text{Mary}, \text{are}, \text{from}, \text{New}, \text{York}]$ and a labeled entity set $\mathbf{y} = \{(1, 1, \text{PER})\}$, unlabeled entity problem is that some entities, like $(6, 7, \text{LOC})$, are neglected by annotators. These unlabeled entities are denoted as $\widetilde{\mathbf{y}} = \{(3, 3, \text{PER}), (6, 7, \text{LOC})\}$.

### 2.2 BERT TAGGING

BERT Tagging is present in Devlin et al. (2019), which adopts IOB tagging scheme, where each token $x_i$ in a sentence $\mathbf{x}$ is labeled with a fine-grained tag, such as B-ORG, I-LOC, or O. Formally, its output is a $n$-length label sequence $\mathbf{z} = [z_1, z_2, \cdots, z_n]$.

Formally, BERT tagging firstly uses BERT to get the representation $\mathbf{h}_i$ for every token $x_i$:

$$[\mathbf{h}_1, \mathbf{h}_2, \cdots, \mathbf{h}_n] = \text{BERT}(\mathbf{x}). \tag{1}$$

Then, the label distribution $\mathbf{q}_i$ is computed as $\text{Softmax}(\mathbf{W}\mathbf{h}_i)$. In training, the loss is induced as $\sum_{1 \leq i \leq n} -\log \mathbf{q}_i[z_i]$. At test time, it obtains the label for each token $x_i$ by $\arg\max \mathbf{q}_i$.

## 3 EMPIRICAL STUDIES

To understand the impacts of *unlabeled entity problem*, we conduct empirical studies over multiple synthetic datasets, different methods, and various metrics.

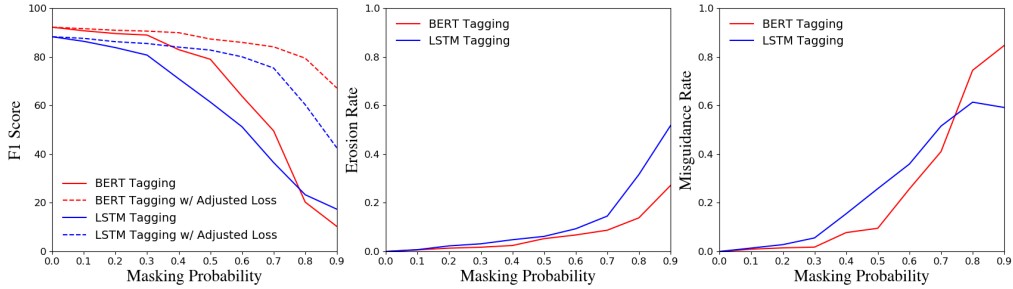

Figure 1: The empirical studies conducted on CoNLL-2003 dataset.

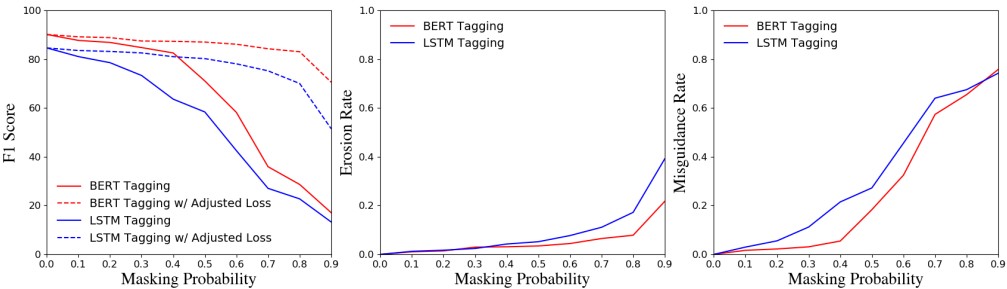

Figure 2: The empirical studies investigated on OntoNotes 5.0 dataset.

### 3.1 PREPARATIONS

**Synthetic Datasets.** We use synthetic datasets to simulate poorly-annotated datasets that contain unlabeled entities. They are obtained by randomly removing the labeled entities of well-annotated datasets with different masking probabilities $p$. The material datasets are CoNLL-2003 (Sang & De Meulder, 2003) and OntoNotes 5.0 (Pradhan et al., 2013). The probabilities $p$ are respectively set as $0.0, 0.1, 0.2, \cdots, 0.9$. In this way, $2 \times 10$ synthetic datasets are constructed.

**Methods.** We adopt two models. One of them is BERT Tagging, which has long been regarded as a strong baseline. The other is LSTM Tagging that replaces the original encoder (i.e., BERT) of BERT Tagging with LSTM (Hochreiter & Schmidhuber, 1997). We use it to study the effect of using pretraining language model. To explore the negative impact brought by unlabeled entities in training, we present an adjusted training loss for above two models:

$$\Big( \sum_{1 \leq i \leq n} - \log \mathbf{q}_i[z_i] \Big) - \Big( \sum_{(i',j',l') \in \widetilde{\mathbf{y}}} \sum_{i' \leq k \leq j'} - \log \mathbf{q}_k[z_k] \Big). \tag{2}$$

The idea here is to remove the incorrect loss incurred by unlabeled entities. Note that missed entity set $\widetilde{\mathbf{y}}$ is reachable in synthetic datasets but unknown in real-world datasets.

**Metrics.** Following prior works, the F1 scores of models are tested by using conlleval script[2]. We also design two degradation rates to measure the different impacts of *unlabeled entity problem*. One is erosion rate $\alpha_p$ and the other is misguidance rate $\beta_p$:

$$\alpha_p = \frac{f_0^a - f_p^a}{f_0^a}, \quad \beta_p = \frac{f_p^a - f_p}{f_p^a}. \tag{3}$$

For a synthetic dataset with the masking probability being $p$, $f_p$ and $f_p^a$ are the F1 scores of a model and its adjusted version, respectively. Note that $f_0^\alpha$ corresponds to $p = 0$. Erosion rate $\alpha_p$ measures how severely the reduction of annotated entities degrades the F1 scores of a model. Misguidance rate $\beta_p$ measures how seriously unlabeled entities misguide the model in training.

---

[2]https://www.clips.uantwerpen.be/conll2000/chunking/conlleval.txt.

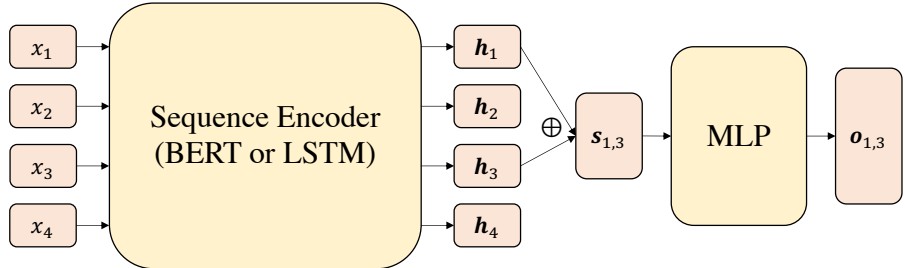

Figure 3: This demonstrates how our model scores possible entities.

## 3.2 OVERALL ANALYSIS

The left parts of Fig. 1 and Fig. 2 show the results of empirical studies, where we evaluate the F1 scores of BERT Tagging and LSTM Tagging on 20 synthetic datasets. From them, we can draw the following observations. Firstly, the significant downward trends of solid lines confirm the fact that NER models severely suffer from *unlabeled entity problem*. For example, by setting the masking probability as $0.4$, the performance of LSTM Tagging decreases by $33.01\%$ on CoNLL-2003 and $19.58\%$ on OntoNotes 5.0. Secondly, in contrast, the dashed lines change very slowly, indicating that the models with adjusted training loss (see Eq. (2)) are much less influenced by the issue. For instance, when masking probability is $0.7$, adopting adjusted loss preserves the F1 scores of BERT Tagging by $41.04\%$ on CoNLL-2003 and $57.38\%$ on OntoNotes 5.0. Lastly, for high masking probabilities, even though the negative impact of unlabeled entities is eliminated by adjusting the training loss, the performance still declines to a certain extent. For example, when masking probability is set as $0.8$, the F1 scores of adjusted LSTM Tagging decrease by $31.64\%$ on CoNLL-2003 and $17.22\%$ on OntoNotes 5.0.

**Reduction of Annotated Entities.** From the last observation, we can infer that a cause of performance degradation is the reduction of annotated entities. In the middle parts of Fig. 1 and Fig. 2, we plot the change of erosion rates $\alpha_p$ (see Eq. (2)) with respect to masking probabilities. We can see that its impact is not very serious when in low masking probabilities but can't be neglected when in high ones. Besides, using pre-training language models greatly mitigates the issue. As an example, when the probability is $0.8$, on both CoNLL-2003 and OntoNotes 5.0, the erosion rates of adjusted BERT Tagging are only about half of those of adjusted LSTM Tagging.

**Misguidance of Unlabeled Entities.** From the last two observations, we can conclude that the primary cause is treating unlabeled entities as negative instances, which severely misleads the models during training. To better understand it, in the right parts of Fig. 1 and Fig. 2, we plot the change of misguidance rates $\beta_p$ (see Eq. (3)) with masking probabilities. These rates are essentially the percentage decreases of F1 scores. From them, we can see that the impact of misguidance is very much serious even when in low masking probabilities.

## 4 METHODOLOGY

Motivated by Sec. 3.2, we present a model that is robust to unlabeled entities.

### 4.1 SCORING MODEL WITH BERT

Based on the findings in Sec. 3.2, we use BERT as the default encoder to mitigate the reduction of annotated entities. Specifically, given a sentence $\mathbf{x}$, we firstly obtain the token representations $\mathbf{h}_i$ with Eq. (1). Then, we get the representation for every phrase $\mathbf{x}_{i,j}$ as

$$\mathbf{s}_{i,j} = \mathbf{h}_i \oplus \mathbf{h}_j \oplus (\mathbf{h}_i - \mathbf{h}_j) \oplus (\mathbf{h}_i \odot \mathbf{h}_j), \tag{4}$$

where $\oplus$ is column-wise vector concatenation and $\odot$ is element-wise vector product. The design here is mainly inspired by Chen et al. (2017).

Finally, multi-layer perceptron (MLP) computes the label distribution $\mathbf{o}_{i,j}$ for a span $(i, j)$:

$$\mathbf{o}_{i,j} = \text{Softmax}(\mathbf{U} \tanh(\mathbf{V}\mathbf{s}_{i,j})). \tag{5}$$

The term $\mathbf{o}_{i,j}[l]$ is the predicted score for an entity $(i, j, l)$.

### 4.2 Training via Negative Sampling

From Sec. 3.2, we know that regarding all the unlabeled spans as negative instances certainly degrades the performances of models, since some of them may be missed entities. Our solution to this issue is negative sampling. Specifically, we randomly sample a small subset of unlabeled spans as the negative instances to induce the training loss.

Given the annotated entity set $\mathbf{y}$, we firstly get all the negative instance candidates as

$$\{(i, j, \text{O}) \mid (i, j, l) \notin \mathbf{y}, 1 \le i \le j \le n, l \in \mathcal{L}\}, \tag{6}$$

where $\mathcal{L}$ is the label space and O is the label for non-entity spans.

Then, we uniformly sample a subset $\widehat{\mathbf{y}}$ from the whole candidate set. The size of sample set $\widehat{\mathbf{y}}$ is $\lceil \lambda * n \rceil, 0 < \lambda < 1$, where $\lceil \rceil$ is the ceiling function.

Ultimately, a span-level cross entropy loss used for training is incurred as

$$\Big( \sum_{(i,j,l) \in \mathbf{y}} - \log(\mathbf{o}_{i,j}[l]) \Big) + \Big( \sum_{(i',j',l') \in \widehat{\mathbf{y}}} - \log(\mathbf{o}_{i',j'}[l']) \Big). \tag{7}$$

Negative sampling incorporates some randomness into the training loss, which reduces the risk of training a NER model with unlabeled entities.

### 4.3 Inference

At test time, firstly, the label for every span $(i, j)$ is obtained by $\arg\max_l \mathbf{o}_{i,j}[l]$. Then, we select the ones whose label $l$ is not O as predicted entities. When the spans of inferred entities intersects, we preserve the one with the highest predicted score and discard the others.

The time complexity of inference is $\mathcal{O}(n^2)$, which is majorly contributed by the span selection procedure. While this seems a bit higher compared with our counterparts, we find that, in practical use, its running time is far less than that of the forward computation of neural networks. The algorithm for inference is greedy yet effective. In experiments, we find that the probability of our heuristic selecting a wrong labeled span when resolving the span conflict is very low.

## 5 Discussion

We show that, through negative sampling, the probability of not treating a specific missed entity in a $n$-length sentence as the negative instance is larger than $1 - \frac{2}{n-3}$:

$$\prod_{0 \le i < \lceil \lambda n \rceil} \Big( 1 - \frac{1}{\frac{n(n+1)}{2} - m - i} \Big) > \Big( 1 - \frac{1}{\frac{n(n+1)}{2} - m - \lceil \lambda n \rceil} \Big)^{\lceil \lambda n \rceil}$$
$$> \Big( 1 - \frac{1}{\frac{n(n+1)}{2} - n - n} \Big)^n \ge \Big( 1 - n * \frac{1}{\frac{n(n+1)}{2} - n - n} \Big) = 1 - \frac{2}{n-3}. \tag{8}$$

Here $\frac{n(n+1)}{2} - m$ is the amount of negative candidates. Besides, we use the facts, $\lambda < 1$, $m \le n$, and $(1 - z)^n \ge 1 - nz, 0 \le z \le 1$, during the derivation.

Note that the above bound is only applicable to the special case where there is just one unlabeled entity in a sentence. We remain the strict proof for general cases to future work.

## 6 Experiments

We have conducted extensive experiments on multiple datasets to verify the effectiveness of our method. Studies on synthetic datasets show that our model can almost eliminate the misguidance

| Masking Prob. | CoNLL-2003 | | OntoNotes 5.0 | |
|---|---|---|---|---|
| | BERT Tagging | Our Model | BERT Tagging | Our Model |
| 0.1 | 90.71 | **91.37** | 87.69 | **89.20** |
| 0.2 | 89.57 | **91.25** | 86.86 | **89.15** |
| 0.3 | 88.95 | **90.53** | 84.75 | **88.73** |
| 0.4 | 82.94 | **89.73** | 82.55 | **88.20** |
| 0.5 | 78.99 | **89.22** | 71.07 | **88.17** |
| 0.6 | 63.84 | **87.65** | 58.17 | **87.53** |

Table 1: The experiment results on two synthetic datasets.

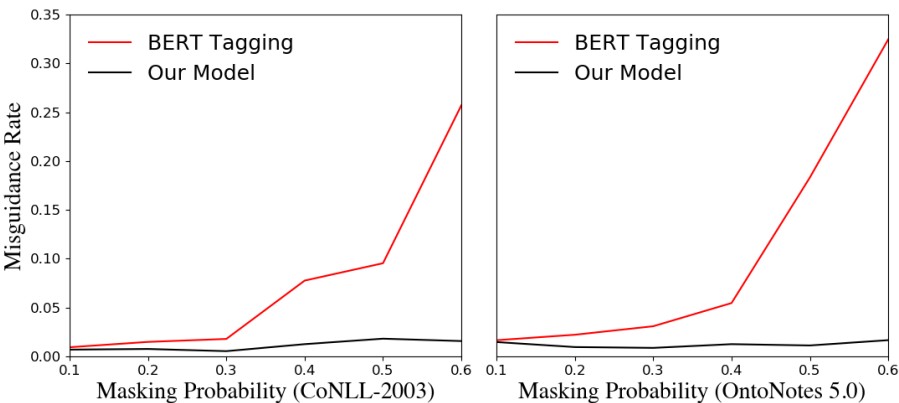

Figure 4: The misguidance rates of BERT Tagging and our model.

brought by unlabeled entities in training. On real-world datasets, our model has notably outperformed prior baselines and achieved the state-of-the-art performances. On well-annotated datasets, our model is competitive to current state-of-the-art method.

## 6.1 SETTINGS

The setup of synthetic datasets and well-annotated datasets are the same as what we describe in Sec. 3.1. For real-world datasets, we use EC and NEWS, both of which are collected by Yang et al. (2018). EC is in e-commerce domain, which has 5 entity types: Brand, Product, Model, Material, and Specification. The data contains 2400 sentences tagged by human annotators and are divided into three parts: 1200 for training, 400 for dev, and 800 for testing. Yang et al. (2018) also construct an entity dictionary of size 927 and apply distant supervision on a raw corpus to obtain additional 2500 sentences for training. NEWS is from MSRA dataset (Levow, 2006). Yang et al. (2018) only adopt the PERSON entity type. Training data of size 3000, dev data of size 3328, and testing data of size 3186 are all sampled from MSRA. They collect an entity dictionary of size 71664 and perform distant supervision on the rest data to obtain extra 3722 training cases by using the dictionary. Both EC and NEWS contain a large amount of incompletely annotated sentences, and hence naturally suffer from the *unlabeled entity problem*.

We adopt the same hyper-parameter configurations of neural networks for all the datasets. L2 regularization and dropout ratio are respectively set as $1 \times 10^{-5}$ and $0.4$ for reducing overfit. The dimension of scoring layers is 256. Ratio $\lambda$ is set as 0.35. When the sentence encoder is LSTM, we set the hidden dimension as 512 and use pretrained word embeddings (Pennington et al., 2014; Song et al., 2018) to initialize word representations. We utilize Adam (Kingma & Ba, 2014) as the optimization algorithm and adopt the suggested hyper-parameters. At evaluation time, we convert the predictions of our models into IOB format and use conlleval script to compute the F1 score. In all the experiments, the improvements of our models over the baselines are statistically significant with rejection probabilities smaller than 0.05.

| Method | CoNLL-2003 | OntoNotes 5.0 |
|---|---|---|
| Flair Embedding (Akbik et al., 2018) | 93.09 | 89.3 |
| BERT-MRC (Li et al., 2020a) | 93.04 | 91.11 |
| HCR w/ BERT (Luo et al., 2020) | 93.37 | 90.30 |
| BERT-Biaffine Model (Yu et al., 2020) | **93.5** | **91.3** |
| Our Model | 93.42 | 90.59 |

Table 2: The experiment results on two well-annotated datasets.

| Method | | EC | NEWS |
|---|---|---|---|
| | String Matching via Ontology | 44.02 | 47.75 |
| | BiLSTM + CRF | 54.59 | 69.09 |
| Yang et al. (2018) | BiLSTM + CRF w/ RL | 56.23 | 73.19 |
| | BiLSTM + Partial CRF | 60.08 | 78.38 |
| | BiLSTM + Partial CRF w/ RL | 61.45 | 79.22 |
| Jie et al. (2019) | Weighted Partial CRF | 61.75 | 78.64 |
| Nooralahzadeh et al. (2019) | BiLSTM + Partial CRF w/ RL | 63.56 | 80.04 |
| This Work | Our Model | **66.17** | **85.39** |
| | Our Model w/o BERT, w/ BiLSTM | **64.68** | **82.11** |

Table 3: The experiment results on two real-world datasets.

## 6.2 RESULTS ON SYNTHETIC DATASETS

In this section, our model is compared with BERT Tagging on the synthetic datasets of the masking probabilities being $0.1, 0.2, \cdots, 0.6$. From Table 1, we can get two conclusions. Firstly, our model significantly outperforms BERT Tagging, especially in high masking probabilities. For example, on CoNLL-2003, our F1 scores outnumber those of BERT Tagging by $1.88\%$ when the probability is $0.2$ and $27.16\%$ when the probability is $0.6$. Secondly, our model is very robust to the *unlabeled entity problem*. When increasing the masking probability from $0.1$ to $0.5$, the results of our model only decrease by $2.35\%$ on CoNLL-2003 and $1.91\%$ on OntoNotes 5.0.

Fig. 4 demonstrates the misguidance rate comparisons between BERT Tagging and our models. The way to adjust our model is reformulating Eq. (7) by defining the negative term via $\{(i, j, \mathrm{O}) \mid \forall l : (i, j, l) \notin \mathbf{y} \cup \widetilde{\mathbf{y}}\}$ rather than the negatively sampled $\hat{\mathbf{y}}$. The idea here is to avoid the unlabeled entities being sampled. From Fig. 4, we can discover that, in all masking probabilities, the misguidance rates of our model are far smaller than those of BERT Tagging and are consistently lower than $2.50\%$. These indicate that, in training, our model indeed eliminates the misguidance brought by unlabeled entities to some extent.

## 6.3 RESULTS ON FULLY ANNOTATED DATASETS

We additionally apply our model with negative sampling on the well-annotated datasets where the issue of incomplete entity annotation is not serious. As shown in Table 2, the F1 scores of our model are very close to current best results. Our model slightly underperforms BERT-Biaffine Model by only $0.09\%$ on CoNLL-2003 and $0.78\%$ on OntoNotes 5.0. Besides, our model surpasses many other strong baselines. On OntoNotes 5.0, our model outperforms HCR w/ BERT by $0.32\%$ and Flair Embedding by $1.44\%$. On CoNLL-2003, the improvements of F1 scores are $0.41\%$ over BERT-MRC and $0.35\%$ over Flair Embedding. All these results indicate that our model is still very effective when applied to high-quality data.

## 6.4 RESULTS ON REAL-WORLD DATASETS

For two real-world datasets, a large portion of training data is obtained via distant supervision. As stated in Yang et al. (2018), the F1 scores of string matching through an entity dictionary are notably declined in terms of the low recall scores, although its precision scores are higher than those of other methods. Therefore, *unlabeled entity problem* is serious in the datasets. As shown in Table 3, the baselines come from three works (Yang et al., 2018; Nooralahzadeh et al., 2019; Jie et al.,

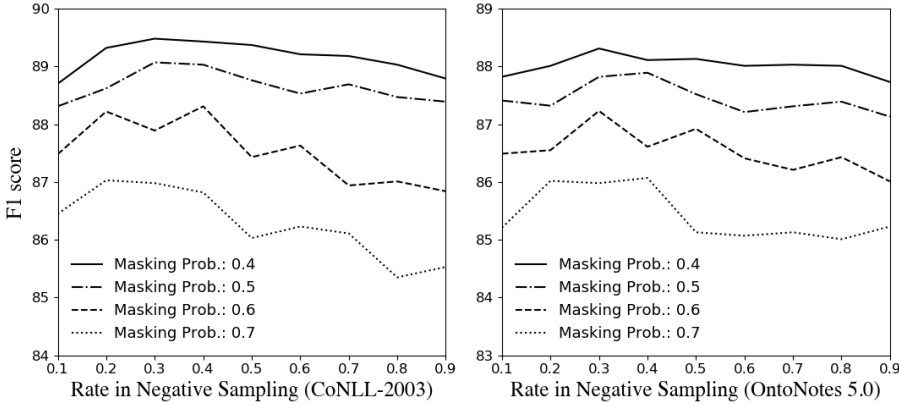

Figure 5: The results of our models with different ratio $\lambda$ on synthetic datasets.

| Metric | CoNLL-2003 | | OntoNotes 5.0 | |
|---|---|---|---|---|
| | BERT Tagging | LSTM Tagging | BERT Tagging | LSTM Tagging |
| Erosion Rate $\alpha_p$ | $-0.94$ | $-0.90$ | $-0.85$ | $-0.82$ |
| Misguidance Rate $\beta_p$ | $-1.00$ | $-0.96$ | $-1.00$ | $-0.98$ |

Table 4: The PCCs between F1 score and degradation rates.

2019). Yang et al. (2018) use Partial CRF to circumvent all possible unlabeled entities and utilize reinforcement learning (RL) to adaptively skip noisy annotation. Jie et al. (2019) and Nooralahzadeh et al. (2019) respectively improve Partial CRF and the policy of RL. All the F1 scores of baselines are copied from Yang et al. (2018); Nooralahzadeh et al. (2019), except for that of Weighted Partial CRF, which is obtained by rerunning its open-source code[3].

Our model has significantly outperformed prior baselines and obtained new state-of-the-art results. Compared with prior best model (Nooralahzadeh et al., 2019), we achieve the improvements of $3.94\%$ on EC and $6.27\%$ on NEWS. Compared with strong baseline, BiLSTM + Partial CRF, the increases of F1 scores are $9.20\%$ and $8.21\%$. To make fair comparisons, we replace BERT with LSTM. Even so, we still outperform (Nooralahzadeh et al., 2019) by $1.76\%$ on EC and $2.52\%$ on NEWS. All these strongly confirm the effectiveness of our model.

### 6.5 Ratio $\lambda$ in Negative Sampling

Intuitively, setting ratio $\lambda$ (see Sec. 4.2) as too large values or too small values both are inappropriate. Large ratios increase the risks of training negatives containing unlabeled entities. Small ratios reduce the number of negative instances used for training, leading to underfitting. Fig. 5 shows the experiments on some synthetic datasets with the ratio $\lambda$ of our method being $0.1, 0.2, \cdots, 0.9$. From it, we can see that all the score curves are roughly arched, which verifies our intuition. Besides, we find that $0.3 < \lambda < 0.4$ performs well in all the cases.

### 6.6 Validity of Degradation Rates

As Table 4 shows, we use Pearson's correlation coefficient (PCC) to measure the statistical correlations between degradation rates (e.g., misguidance rate $\beta_p$) and the F1 score. We can see that the correlation scores are generally close to $-1$. For example, for LSTM Tagging, on the synthetic datasets built from CoNLL-2003, the correlation score of erosion rate $\alpha_p$ is $-0.90$ and that of misguidance rate $\beta_p$ is $-0.96$. The results indicate that not only degradation rates quantify specific impacts of unlabeled entities but also their negative values change synchronously with the F1 score. We conclude that degradation rates are appropriate metrics for evaluation.

---

[3]https://github.com/allanj/ner_incomplete_annotation.

## 7 RELATED WORK

NER is a classical task in information extraction. Previous works commonly treat it as a sequence labeling problem by using IOB tagging scheme (Huang et al., 2015; Akbik et al., 2018; Luo et al., 2020; Li et al., 2020b;c). Each word in the sentence is labeled as B-tag if it is the beginning of an entity, I-tag if it's inside but not the first one within the entity, or O otherwise. This approach is extensively studied in prior works. For example, Akbik et al. (2018) propose Flair Embedding that pretrains character embedding in large corpora and uses it rather than token representations to represent a sentence. Recently, there is a growing interest in span-based models (Li et al., 2020a; Yu et al., 2020). They treat the spans, instead of single words, as the basic units for labeling. For example, Li et al. (2020a) present BERT-MRC that regards NER as a MRC task, where named entities are extracted as retrieving answer spans. Span-based models are also prevalent in language modeling (Li et al., 2020d), syntactic analysis (Stern et al., 2017), etc.

In some practical applications (e.g., fine-grained NER (Zhang et al., 2020)), NER models are faced with *unlabeled entity problem*, where the unlabeled entities seriously degrade the performances of models. Several approaches to this issue have been proposed. Fuzzy CRF and AutoNER (Shang et al., 2018b) allow learning from high-quality phrases. However, since these phrases are obtained through distant supervision, the unlabeled entities in the corpora may still be missed. PU learning (Peng et al., 2019; Mayhew et al., 2019) unbiasedly and consistently estimates the training loss. Nevertheless, the unlabeled entities still impact the classifiers of the corresponding entity types and, importantly, the model can't disambiguate neighboring entities. Partial CRF (Yang et al., 2018; Jie et al., 2019) supports learning from incomplete annotations. However, because fully annotated corpora are still needed to training models with true negative instances, this type of approach is not applicable to the situations where no high-quality data is available.

## 8 CONCLUSION

In this work, we study what are the impacts of unlabeled entities on NER models and how to effectively eliminate them. Through empirical studies performed on synthetic datasets, we find two causes: the reduction of annotated entities and treating unlabeled entities as training negatives. The first cause has fewer influences than the second one and can be mitigated by adopting pretraining language models. The second cause seriously misleads the models in training and greatly affects their performances. Based on the above observations, we propose a novel method that is capable of eliminating the misguidance of unlabeled entities during training. The core idea is to apply negative sampling that avoids training NER models with unlabeled entities. Experiments on synthetic datasets and real-world datasets demonstrate that our model handles unlabeled entities well and significantly outperforms previous baselines. On well-annotated datasets, our model is competitive with the existing state-of-the-art approach.

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
