# OpenReview forum: "Empirical Analysis of Unlabeled Entity Problem in Named Entity Recognition"
_ICLR.cc/2021/Conference — ICLR 2021 Poster_

### Official Review · AnonReviewer2 · 2020-10-19
**Practical, interesting results**

**Rating:** 7
**Confidence:** 4

**Review:**

This paper conducted an empirical analysis on the unlabeled entity problem in the NER task. It concluded that there are two reasons to affect the NER model's performance:  the reduction of annotated entities, and treating unlabeled entities as negative instances. Experiments showed that the latter reason gave a much more negative impact on the NER models.

This empirical study was conducted on the synthetic datasets which were extracted from two English NER datasets using negative sampling. The way of constructing synthetic datasets should be explained clearly (Section 3.1). It is not clear how does the negative sampling applied on the dataset (in sentence level? entity level? token level?).

---

> ### Author Response · Authors · 2020-11-17
> **Response to AnonReview2**
>
> Thanks for your valuable feedbacks.
>
> Comment-1: The way of constructing synthetic datasets should be explained clearly (Section 3.1).
>
> Answer-1: Synthetic datasets help us understand how unlabeled entity problem impacts NER models. A synthetic dataset is constructed from a well-annotated dataset (e.g., CoNLL-2003) as follows. Given a probability p (say p=0.3), for each annotated entity in a sentence from a well-annotated dataset, we randomly sample a number \mu (0<\mu<1) from the uniform distribution. If \mu <= 0.3, then we drop this entity for the underlying sentence; otherwise we retain this entity. We will explain it more clearly in the revised version.
>
> Comment-2: It is not clear how does the negative sampling applied on the dataset (in sentence level? entity level? token level?).
>
> Answer-2: Given a training case (i.e., a sentence and annotated entities), negative sampling is to randomly sample a subset of unlabeled spans (that are not the spans of any annotated entities) as the negative instances for training. Therefore, negative sampling is applied at entity level (more precisely, span level). Eq. (6) and Eq. (7) also indicate this fact. We will make it more clear in the revised version.

---

### Official Review · AnonReviewer4 · 2020-10-27
**A plausible approach but a weak comparison**

**Rating:** 6
**Confidence:** 4

**Review:**

This paper focuses on the unlabeled entity problem in NER, where the entities of a sentence are incomplete annotated. Since some entities may not be annotated, the performance of models can be degraded. This paper analyzes the performance degradation by evaluating synthetic datasets and finds that all the unlabeled entities are treated as negative instances is the main factor of the performance degradation. To alleviate the performance degradation, this paper proposes a negative sampling approach that considers only a small subset of unlabeled entities in order to reduce the impacts of unlabeled entities. The experimental results show that the proposed method achieves better performances compared to previous studies on real-world datasets and achieves competitive performances compared to the state-of-the-art methods on well-annotated datasets.

Strong points:
+ The analysis using two proposed metrics to find causes of the performance degradation of the unlabeled entity problem is quite interesting.
+ The idea of using the negative sampling under the data imbalance situation, especially in NER, seems reasonable.
+ The paper is well-written and easy to follow. It provides comprehensive experiments on several datasets.

Concerns:
1. The key concern about this paper is the missing of some references. There are some previous studies related to the unlabeled entity problem. Particularity, this problem is quite related to or almost the same as the “Incomplete annotations” at [1] or “Partially annotated training data” at [2]. In addition, the unlabeled entity problem has a little relevance to the data imbalances [3] if we assume that all unlabeled spans are negative instances. Therefore, the proposed method should be compared with these previous studies experimentally and/or theoretically.
[1] Jie et al., Better Modeling of Incomplete Annotations for Named Entity Recognition, NAACL 2019.
[2] Mayhew et al., Named Entity Recognition with Partially Annotated Training Data, CONLL 2019.
[3] Li et al., Dice Loss for Data-imbalanced NLP Tasks, ACL 2020.
2. In the experiments on the real-word dataset, the proposed method only compared with weak baselines. The reviewer suggests the authors to evaluate with the state-of-the-art NER methods such as BERT-MRC or BERT-Biaffine Model.
3. At the inference time, every span should be input to the MLP layer to obtain the predicted score. That is, the proposed method takes more time (O(n^2)) to infer new sentences compared to other methods (O(n)). Furthermore, this paper uses heuristic(s) when the spans of inferred entities intersects. However, there is no discussion about them.

Minor comments:
1. In the section 3.2, the figures and the contents about the figures are inconsistent. That is, the middle parts of the figures are for the change in misguidance rates and the right-hand ones are for the change of the erosion rates.
2. For batching with sampling, it would be better to provide more details about it.
3. It would be nice to show a model convergence or loss convergence graph because this paper uses random sampling.
4. It would be better to show the correlation in numbers between the f1 scores and the proposed rates.

---

> ### Author Response · Authors · 2020-11-17
> **Response to AnonReview4 (First Three Comments)**
>
> Thanks for your valuable comments.
>
> Comment-1: The key concern about this paper is the missing of some references. There are some previous studies related to the unlabeled entity problem. Particularity, this problem is quite related to or almost the same as the “Incomplete annotations” at [1] or “Partially annotated training data” at [2]. In addition, the unlabeled entity problem has a little relevance to the data imbalances [3] if we assume that all unlabeled spans are negative instances. Therefore, the proposed method should be compared with these previous studies experimentally and/or theoretically. [1] Jie et al., Better Modeling of Incomplete Annotations for Named Entity Recognition, NAACL 2019. [2] Mayhew et al., Named Entity Recognition with Partially Annotated Training Data, CONLL 2019. [3] Li et al., Dice Loss for Data-imbalanced NLP Tasks, ACL 2020.
>
> Answer-1: We are sorry for the missing references. [1] further optimizes Partial CRF [4] and [2] is based on PU Learning [5]. In fact, both [4] and [5] are cited in our paper. Partial CRF demands well-annotated data to be trained with true negative instances and PU Learning relies on prior information and heuristics. They both lack enough flexibility compared with our method. To make comparisons on performances, we run the open source codes of [1] (https://github.com/allanj/ner_incomplete_annotation) and [5] (https://github.com/v-mipeng/LexiconNER) on the two real-world datasets (Note that [4] is already compared in our paper and the source code of [2] is not available). The results are shown in the table below.
>
> Method                                                                 EC                     NEWS
>
> Partial CRF (Jie et al., 2019)                               61.75                   78.64
>
> PU Learning                                                        61.22                  77.98
>
> Our Model w/o BERT w/ BiLSTM                    64.68                  82.11
>
> From the table, we can see that our proposed method has notably outperformed both Partial CRF and PU Learning. The improvements over both baselines are at least 4.74% on EC and 4.41% on NEWS. Your comment about data imbalances is very interesting and we will explore it in future work. We will add the experiment results and discussions above in the revised version.
>
> [1] Jie et al., Better Modeling of Incomplete Annotations for Named Entity Recognition, NAACL-2019
>
> [2] Mayhew et al., Named Entity Recognition with Partially Annotated Training Data, CONLL-2019
>
> [3] Li et al., Dice Loss for Data-imbalanced NLP Tasks, ACL 2020
>
> [4] Yang et al., Distantly supervised ner with partial annotation learning and reinforcement learning, COLING-2018
>
> [5] Peng et al., Distantly Supervised Named Entity Recognition using Positive-Unlabeled Learning, ACL-2019
>
> Comment-2: In the experiments on the real-word dataset, the proposed method only compared with weak baselines. The reviewer suggests the authors to evaluate with the state-of-the-art NER methods such as BERT-MRC or BERT-Biaffine Model.
>
> Answer-2: According to your suggestion, we additionally run three models (BERT-MRC, BERT-Biaffine, and BERT + Partial CRF) on the two real-world datasets. The results are demonstrated in the table below.
>
> Method                                           EC                    NEWS
>
> BERT-MRC                                    55.72                  74.55
>
> BERT-Biaffine                               55.99                  74.57
>
> BERT + Partial CRF                      63.12                   81.24
>
> Our Model                                    66.17                  85.39
>
> We will add these experiment results in the revised version.
>
> Comment-3: At the inference time, every span should be input to the MLP layer to obtain the predicted score. That is, the proposed method takes more time (O(n^2)) to infer new sentences compared to other methods (O(n)). Furthermore, this paper uses heuristic(s) when the spans of inferred entities intersects. However, there is no discussion about them.
>
> Answer-3: The theoretical time complexity O(n^2) is due to the span selection procedure of our model. In fact, its running time is much less than that of the forward computation of neural networks (even for LSTM). In experiments, we find the latter is 10 times longer than the former. The heuristic used to solve span conflicts is simple yet effective. Empirically, the probability of our strategy making mistakes (the labeled span with higher score is not the true entity) is lower than 0.07%. We will clarify these facts in the revised version of our paper.

---

> > ### Author Response · Authors · 2020-11-17
> > **Response to AnonReview4 (Last Four Comments)**
> >
> > Comment-4: In the section 3.2, the figures and the contents about the figures are inconsistent. That is, the middle parts of the figures are for the change in misguidance rates and the right-hand ones are for the change of the erosion rates.
> >
> > Answer-4: we will rearrange the figures so that they are consistent with the contents.
> >
> > Comment-5: For batching with sampling, it would be better to provide more details about it.
> >
> > Answer-5: We will provide more details in the revised version and make detailed comments in the released source code.
> >
> > Comment-6: It would be nice to show a model convergence or loss convergence graph because this paper uses random sampling.
> >
> > Answer-6: We will add them in the revised version.
> >
> > Comment-7: It would be better to show the correlation in numbers between the f1 scores and the proposed rates.
> >
> > Answer-7: According to your suggestion, we calculate the Pearson Correlation Coefficient for both baselines (i.e., BERT Tagging and LSTM-tagging). The results are in the table below.
> >
> >
> > 1. Result for BERT Tagging
> >
> >                                                Erosion Rate              Misguidance Rate
> >
> >
> > CoNLL-2003                            -0.94                                 -1.00
> >
> > OntoNotes 5.0                         -0.85                                 -1.00
> >
> >
> > 2. Result for LSTM Tagging
> >
> >                                                Erosion Rate              Misguidance Rate
> >
> > CoNLL-2003                            -0.90                                -0.96
> >
> > OntoNotes 5.0                         -0.82                                -0.98
> >
> > We will add these results in the revised version.

---

### Official Review · AnonReviewer3 · 2020-10-28
**Generally good, a few questions**

**Rating:** 5
**Confidence:** 4

**Review:**

This paper investigates the unlabeled entity problem, which is generally observed in the manual annotation setting and distant supervision as well. The unlabeled problem is important and some existing works focus on solving the problems using partial CRF setting or data selector. The main observation of this paper lies in two aspects: 1) comparison between the reduction of annotated entities or treating unlabeled entities as negative instances.  Most interestingly, the authors show the observed difference between pre-trained language models and LSTM-based models.  Based on the observations, they propose a general approach to eliminate the misguidance brought by unlabeled entities and such a simple design shows good performances.

The Paper is overall well written and easy to follow. But  I still have a few questions and want to get answers from authors.

Questions:

1) The first question is about 4.2 Training via Negative sampling on page 5. I am not quite sure about the procedure. Negative instance candidates are randomly selected from original sentences.  You use \hat{y}, which is a subset of randomly selected span to replace a missed entity set defined in Eq. (2)?

2) Could you expand more about Equation 8 to add more details?

3) The unlabeled entity problem is most serious in the distant supervision setting. However, the distant supervision setting suffers from entity ambiguation and unlabeled entity problem simultaneously.  How do you think your design to tackle entity ambiguation problem? Moreover, in the distant supervision experiment in Table  3, how will you model compare with other distant supervision models like AutoNER?

---

> ### Author Response · Authors · 2020-11-17
> **Response to AnonReview3 (First Two Questions)**
>
> Thanks for your valuable comments.
>
> Question-1: The first question is about 4.2 Training via Negative sampling on page 5. I am not quite sure about the procedure. Negative instance candidates are randomly selected from original sentences. You use \hat{y}, which is a subset of randomly selected span to replace a missed entity set defined in Eq. (2)?
>
> Answer-1: We are sorry for the confusion. Actually, negative instance candidates denote all spans except manually labeled entities. Since these negative instance candidates may contain some real entities which are just neglected by annotators, we sample a subset \hat{y} from these candidates as the negative instances for training (see Eq. (7)). \tilde{y} in Eq. (2) is different from \hat{y} in Eq. (7). The former denotes a set of named entities neglected by annotators while the latter is a set of negative instances for training.
>
> For example, x= [Jack is from New York] and y = [(2, 3，LOC)] is the entity set annotated by human. We can see that human neglected an entity (0, 0, PER) and thus \tilde{y} = [(0, 0, PER)].
>
> In this way, negative instance candidates set include all spans expect (2, 3，LOC), i.e., it is [(0, 0, O), (0, 1, O), (0, 2, O), (0, 3, O), (0, 4, O), (1, 1, O), (1, 2, O), (1, 3, O), (1, 4, O), (2, 1, O), (2, 2, O), (2, 4, O), (3, 3, O), (3, 4, O), (4, 4, O)].
>
> In addition, if we sample four negative instances as \hat{y}, then \hat{y} may be [(0, 3, O), (4, 4, O), (1, 1, O), (2, 4, O)].
>
> We will add this example to explain these notations more clearly in our revised paper.
>
> Question-2: Could you expand more about Equation 8 to add more details?
>
> Answer-2: Eq. (8) theoretically proves that, through negative sampling, the probability of treating a neglected entity as the negative instance is very low. To verify this conclusion in real scenarios, for every synthetic dataset, we calculate the percentage of neglected entities (i.e., the entities in \tilde{y}) in the set of sampled negative instances (i.e., \hat{y}). The results are demonstrated in the table below.
>
> Dataset\Masking Prob.        0.1           0.2             0.3           0.4            0.5           0.6          0.7            0.8           0.9
>
> CoNLL-2003                           0.17%      0.41%        0.62%      0.83%       1.12%       1.41%      1.67%       1.66%       1.78%
>
> OntoNotes 5.0                      0.07%      0.15%        0.21%      0.27%       0.35%      0.42%     0.48%       0.56%      0.63%
>
> Note that all the percentage scores are averaged over 10 epochs and the \lambda ratio (see Section 4.2) is set as 0.9. From the table, we can see that, empirically, the possibility of our negative sampling making mistakes (treating a neglected entity as a negative instance) is very low. These results further confirm the effectiveness of our negative sampling. We will add this experiment and make a deep analysis in the revised version.

---

> > ### Author Response · Authors · 2020-11-17
> > **Response to AnonReview3 (Last Two Questions)**
> >
> > Question-3: The unlabeled entity problem is most serious in the distant supervision setting. However, the distant supervision setting suffers from entity ambiguation and unlabeled entity problem simultaneously. How do you think your design to tackle entity ambiguation problem?
> >
> > Answer-3: This paper focuses on analyzing and solving the unlabeled entity problem. Experiments show that our proposed method can almost eliminate the misguidance brought by unlabeled entities during training (see Section 6.2 and Figure 4). As for the entity ambiguation problem, a feasible idea is, in the preprocessing stage, using the context information to disambiguate entity labels.
> >
> > Question-4: Moreover, in the distant supervision experiment in Table 3, how will you model compare with other distant supervision models like AutoNER?
> >
> > Answer-4: Besides Partial CRF, other related methods mainly include AutoNER [1] and PU Learning [2,3]. AutoNER allows learning from high-quality phrases that may be neglected entities. Because obtaining these phrases demands external resources, we can’t offer direct comparisons. Note that our negative sampling doesn’t rely on any external resource, which is more flexible. PU Learning assigns low weights to false negative instances in the loss function. LexiconNER [2] is based on PU Learning and has released its source code at https://github.com/v-mipeng/LexiconNER. We run the code on two real-world datasets (described in Section 6.4) and the results are shown in the table below.
> >
> > Method                                                               EC                    NEWS
> >
> > PU Learning                                                      61.22                  77.98
> >
> > Our Model w/o BERT w/ BiLSTM                 64.68                  82.11
> >
> > From the table, we can see that our proposed model has notably outperformed PU Learning on both datasets. The improvements of F1 scores are respectively 5.65% and 5.30% on EC and NEWS. We will add these results and make a detailed analysis in the revised version.
> >
> > [1] Shang et al., Learning Named Entity Tagger using Domain-Specific Dictionary, EMNLP-2018
> >
> > [2] Peng et al., Distantly Supervised Named Entity Recognition using Positive-Unlabeled Learning, ACL-2019
> >
> > [3] Mayhew et al., Named Entity Recognition with Partially Annotated Training Data, CoNLL-2019

---

### Official Review · AnonReviewer1 · 2020-10-30
**a good idea supported by consistent experiments**

**Rating:** 8
**Confidence:** 4

**Review:**

This article deals with the problem of partially labeled dataset: if some entities are missing, how SOTA approaches are going to behave? To answer this question, the authors degrade CoNLL classical dataset by masking a pourcentage of the labeled data. Then, they wonder which part of the missing performance is due to the lack of labels and which part is due to the incorrect labelling of discarded supervision.
The experiments are well explained and interesting on synthetic dataset. Then the authors propose a new cross entropy loss to test their hypothesis on real data by sampling high confidense negative samples as ground truth. It is a way of performing distillation on the model using negative sampling.

Consistant & relevant work that deserves to be publised in ICLR.

* We wonder what would give a classical distillation process on this task. [even if relevant comparison are made with results from the litterature]

* Given the architecture, we wonder what is the detailed learning procedure: it seems clear that the network is first trained on the real ground truth and then refined using the distillation loss. Section 3.1 is a little bit short on this point.

* Regarding the model, equation (4) is not discussed nor analysed.

* The approach is rather simple but elegant.

---

> ### Author Response · Authors · 2020-11-17
> **Response to AnonReview1**
>
> Thanks for your valuable feedbacks.
>
> Comment-1: We wonder what would give a classical distillation process on this task. [even if relevant comparison are made with results from the litterature]
>
> Answer-1: Mainstream approaches to the problem include AutoNER [1], Partial CRF [2,3], and PU Learning [4,5]. AutoNER supports learning from high-quality phrases that may be unlabeled entities. However, mining these phrases demands external domain-specific corpora, which is not flexible. Partial CRF marginalizes the loss over all possible complete label sequences that are compatible with the incomplete annotation. However, it still demands well-annotated data to be trained with true negative instances. In experiments, our proposed methods have significantly outperformed Partial CRF on both real-world datasets (see Section 6.4 and Table 3). PU Learning assigns low weights to false negative instances in the loss function. However, obtaining these weights relies on prior information and heuristics, which is also inflexible.
>
> A work [4] based on PU Learning has released its source code [https://github.com/v-mipeng/LexiconNER]. We run the code on two real-world datasets to make comparisons with PU Learning. The results are shown in the table below.
>
> Method                                                               EC                     NEWS
>
> PU Learning                                                     61.22                  77.98
>
> Our Model w/o BERT w/ BiLSTM                  64.68                  82.11
>
> We will add these results and make a clear comparison in the revised version.
>
> Comment-2: Given the architecture, we wonder what is the detailed learning procedure: it seems clear that the network is first trained on the real ground truth and then refined using the distillation loss. Section 3.1 is a little bit short on this point.
>
> Answer-2: We didn’t adopt multi-stage training strategies. On every synthetic dataset, we evaluate the performances of models with adjusted loss (see Eq. (2)) and MLE loss, separately. The two performances are used to show how unlabeled entity problem affects NER models. For example, the left part of Figure 1 demonstrates the results of four different settings (BERT Tagging w/ MLE, BERT Tagging w/ Adjusted Loss, LSTM Tagging w/ MLE, and LSTM Tagging w/ Adjusted Loss) on the synthetic datasets derived from CoNLL-2003. Each model is trained from scratch on a dataset. We will make this more clear in the revised version.
>
> Comment-3: Regarding the model, equation (4) is not discussed nor analysed.
>
> Answer-3: The design of Eq. (4) is inspired by ESIM model (see Eq. (14) in [6]). We also tried other options such as bi-affine scoring [7], but we found Eq. (4) performs the best. We will clarify this in the revised version.
>
> [1] Shang et al., Learning Named Entity Tagger using Domain-Specific Dictionary, EMNLP-2018
>
> [2] Yang et al., Distantly supervised ner with partial annotation learning and reinforcement learning, COLING-2018
>
> [3] Jie et al., Better Modeling of Incomplete Annotations for Named Entity Recognition, NAACL-2019
>
> [4] Peng et al., Distantly Supervised Named Entity Recognition using Positive-Unlabeled Learning, ACL-2019
>
> [5] Mayhew et al., Named Entity Recognition with Partially Annotated Training Data, CoNLL-2019
>
> [6] Chen et al., Enhanced LSTM for Natural Language Inference, ACL-2017
>
> [7] Dozat et al., DEEP BIAFFINE ATTENTION FOR NEURAL DEPENDENCY PARSING, ICLR-2017

---

### Decision · Program_Chairs · 2021-01-07
**Final Decision**

**Decision:**

Accept (Poster)

**Comment:**

This paper studies the unlabeled entity problem in NER. Specifically, performance degradation in training of NER models due to unlabeled entities. It analyzes the reason through evaluation on synthetic datasets and finds that it is due to the fact that all the unlabeled entities are treated as negative examples. To cope with the problem, it proposes a negative sampling method which considers the use of only a small subset of unlabeled entities. Experimental results show that the proposed method achieves better performances than the baselines on real-world datasets and achieves competitive performances compared with the state-of-the-art methods on well-annotated datasets.

Pros
•	The paper is clearly written.
•	The proposed method appears to be technically sound.
•	Experimental results support the main claims.
•	The findings in the paper are useful for the field.

Cons
•	Novelty of the work might not be enough.

The authors have addressed some clarity and reference issues pointed out by the reviewers in the rebuttal.  Discussions have been made among the reviewers.